# Iterated Population Based Training with Task-Agnostic Restarts

**Alexander Chebykin** [1]  **Tanja Alderliesten** [2]  **Peter A. N. Bosman** [1][3]

## Abstract

Hyperparameter Optimization (HPO) can lift the burden of tuning hyperparameters (HPs) of neural networks. HPO algorithms from the Population Based Training (PBT) family are efficient thanks to dynamically adjusting HPs every few steps of the weight optimization. Recent results indicate that *the number of steps between HP updates* is an important meta-HP of all PBT variants that can substantially affect their performance. Yet, no method or intuition is available for efficiently setting its value. We introduce Iterated Population Based Training (IPBT), a novel PBT variant that automatically adjusts this HP *via restarts* that reuse weight information in a task-agnostic way and leverage time-varying Bayesian optimization to reinitialize HPs. Evaluation on 8 image classification and reinforcement learning tasks shows that, on average, our algorithm matches or outperforms 5 previous PBT variants and other HPO algorithms (random search, ASHA, SMAC3), without requiring a budget increase or any changes to its HPs. The source code is available online.

## 1. Introduction

Machine learning often achieves excellent results thanks to well-tuned hyperparameters (HPs). Hyperparameter Optimization (HPO) algorithms can reduce the time and effort needed for tuning (Feurer & Hutter, 2019; Tan et al., 2024).

Efficiency is important for HPO in general but it is vital when the underlying task is expensive, as is the case for neural network training, where often only a few training runs can be afforded. One HPO algorithm that is well-suited for this setting is Population Based Training (PBT) (Jaderberg et al., 2017). PBT draws its efficiency from dynamically

adjusting HPs during training. Specifically, a population of networks – each with its own set of HPs – undergoes weight optimization via gradient descent for several steps. After this, poorly performing networks are replaced by copies of better-performing ones, with their corresponding HPs copied and perturbed.

Further efficiency gains can be achieved by replacing random perturbations with Bayesian Optimization (BO), an idea explored by several PBT variants (Parker-Holder et al., 2020; 2021; Wan et al., 2022). Despite the reported improvements, it was recently shown (Chebykin et al., 2025) that none of the PBT variants consistently outperforms others, with substantial variability due to the value of the "step size" HP (i.e., how many weight update steps are performed before HPs are adjusted). The step size varies across papers with no discussion on how it is set for the tasks at hand or how it should be set for unseen tasks. Having to empirically determine a good value for this HP detracts from the utility and efficiency of PBT variants.

We aim to address this issue by having the step size adjusted automatically without sacrificing efficiency. Towards this goal, we introduce Iterated Population Based Training (IPBT), a novel PBT variant that performs several iterations of a BO PBT procedure with a step size that is increased on each restart. A restart is triggered upon hitting diminishing returns, as determined by our novel data-driven mechanism. Crucially for efficiency, information is transferred across restarts in a task-agnostic manner: the partially-trained weights are shrink-perturbed (Ash & Adams, 2020) (i.e., reduced in magnitude and injected with noise), while HPs are reinitialized via a time-varying meta BO procedure that learns across restarts. Figure 1 presents an overview of our algorithm and an example run.

IPBT builds upon the idea of PBT with restarts as introduced in Bayesian Generational PBT (BG-PBT) (Wan et al., 2022). Key differences of IPBT relative to BG-PBT are the step size adjustment upon restarts, *task-agnostic* weight reuse after a restart, and reinitialization of HPs across restarts via time-varying BO. See Sections 2.2 and 3 for further details.

An important advantage of PBT variants is that they run in parallel, taking approximately the same wall-clock time as training a single network. To uphold this advantage, IPBT is designed not to increase the total number of weight update

---

[1]Centrum Wiskunde & Informatica, Amsterdam, the Netherlands [2]Leiden University Medical Center, Leiden, the Netherlands [3]Delft University of Technology, Delft, the Netherlands. Correspondence to: Alexander Chebykin <a.chebykin@cwi.nl>.

*Proceedings of the 43rd International Conference on Machine Learning*, Seoul, South Korea. PMLR 306, 2026. Copyright 2026 by the author(s).

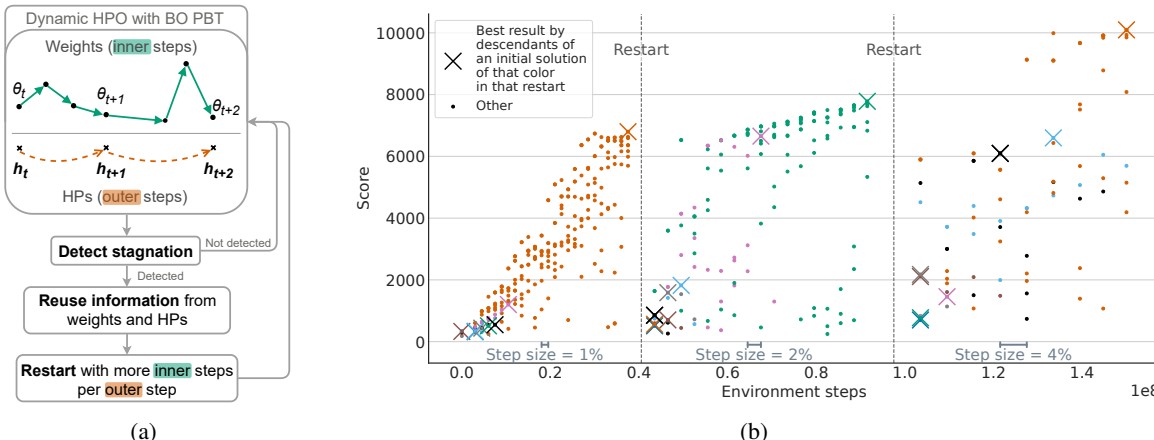

*Figure 1. (a)* Overview of IPBT and *(b)* an example run on the Humanoid task. Within each iteration (three can be seen in *(b)*), HPs are dynamically optimized during training via a BO PBT procedure (Section 2.2). If stagnation of performance is detected (Section 3.2), a restart is triggered. Information from weights and HPs of previous iterations is reused (Section 3.3), and the outer step size of the PBT procedure is increased (Section 3.4). New HPs are predicted via time-varying meta BO trained on the initial HPs of previous iterations as inputs and their descendants' maximum achieved performance as outputs (in *(b)*, the descendants of each initial solution share its color).

steps, restarting aggressively to most effectively utilize the limited budget. We explicitly target low-budget HPO, with the budget in our main experiments set to 8 full training runs. Additionally, we assume that no prior information about the task is available, motivating a black-box HPO approach.

We evaluate IPBT on image classification tasks using CIFAR-10/100 (Krizhevsky & Hinton, 2009), Fashion-MNIST (Xiao et al., 2017), and TinyImageNet (Stanford CS231N, 2017), as well as on reinforcement learning tasks using Brax (Freeman et al., 2021) Humanoid, Hopper, Pusher, Walker. IPBT is compared to 5 previous PBT variants (see Sections 2.2 and 4.1) with untuned and tuned step sizes, as well as Random Search (RS) (Bergstra et al., 2011), Asynchronous Successive Halving Algorithm (ASHA) (Li et al., 2020), and SMAC3 (Lindauer et al., 2022). We additionally perform a thorough ablation study of the components of IPBT to better understand their influence.

Our main contributions are as follows:

- We introduce IPBT, a novel PBT variant that efficiently adjusts its step size via restarts.

- IPBT is evaluated on 8 image classification and reinforcement learning tasks, where it is compared with 5 previous PBT variants and other HPO algorithms (RS, ASHA, SMAC3).

- We explore how different methods of reusing information contained in weights and HPs impact performance.

## 2. Related work

### 2.1. Efficient algorithms for expensive HPO

BO algorithms are a good option for expensive optimization tasks, including HPO (Snoek et al., 2012; Cowen-Rivers et al., 2022). However, since BO algorithms are typically initialized with as many random samples as the problem dimensionality, they face difficulties in low-budget HPO settings, where the budget in terms of full training runs is often smaller than the dimensionality. Therefore, while efficient algorithms for expensive HPO often have BO components, they pursue additional efficiency in other ways.

PBT variants (BO and non-BO) are efficient thanks to dynamic HPO where HPs are changed during training based on short-term feedback (we discuss these algorithms in Section 2.2). Alternatively, the HPs can be fixed during training but the training itself may be shorter, use a smaller model, or rely on some other proxy. These algorithms are called "multi-fidelity", as the proxy task may be parameterized to be more or less expensive (and accordingly provide information with higher or lower fidelity) (Won et al., 2025).

In Successive Halving (SH) (Jamieson & Talwalkar, 2016), $N$ randomly sampled configurations are trained for a number of epochs. After this, all but the best $\frac{1}{\eta}$ configurations are stopped, and the remaining ones continue training for $\eta$ times as many epochs (default $\eta = 2$). This procedure repeats until the maximum per-configuration budget $R$ is reached. SH avoids wasting time on unpromising configurations but suffers from greediness and does not learn from experience. Hyperband (Li et al., 2018) improves upon SH by running multiple instances of it with different values of $(N, R)$. Hyperband still relies on random sampling, which

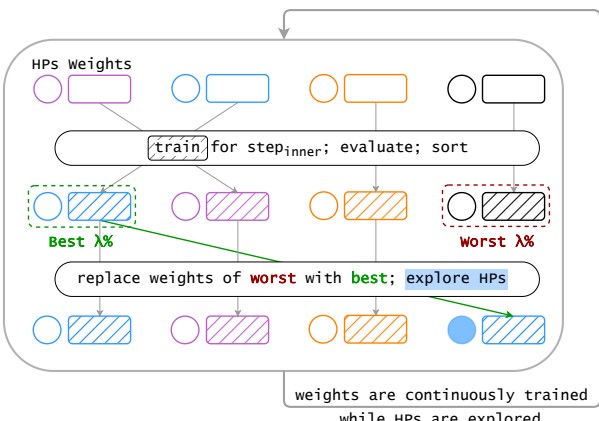

*Figure 2.* General PBT loop: in each outer step, the weights are trained for $\text{step}_{\text{inner}}$ inner steps and the HPs are explored.

was replaced with BO in BO Hyperband (BOHB) (Falkner et al., 2018). Awad et al. (2021) showed that replacing BO with differential evolution further improves results, while Lindauer et al. (2022) demonstrated that using random forest as a predictor in BO performs even better, achieving excellent results as a part of the SMAC3 package. Throughout this paper, we use "SMAC3" to refer to the multi-fidelity setup of this package.

In another research line, ASHA (Li et al., 2020) effectively parallelized SH and was shown to outperform SH, Hyperband, and BOHB without any learning, thus making it the most effective approach based purely on random exploration.

### 2.2. Population Based Training variants

As mentioned previously, PBT (Jaderberg et al., 2017) enables efficient HPO by dynamically adapting HPs while continuously training the weights. The optimization is bilevel, with inner steps updating the weights and outer steps updating the HPs. PBT operates with a population of $N$ solutions where each solution at an outer step $t$ is a pair of HPs and weights, $(\boldsymbol{h}_t^i, \theta_t^i)$. The weights of each solution $\theta_t^i$ are trained using the corresponding HPs $\boldsymbol{h}_t^i$ for $\text{step}_{\text{inner}}$ steps (e.g., via gradient descent), after which the weights are evaluated on a validation set. Each of the worst-performing $\lambda\%$ solutions is replaced with a copy of one of the best-performing $\lambda\%$ solutions (thus ensuring propagation of well-performing weights), and the HPs of the copies are explored via random perturbation. A graphical overview of the PBT loop is shown in Figure 2.

In Population Based Bandits (PB2) (Parker-Holder et al., 2020), BO was suggested as a replacement of random perturbation for exploring HPs. A dataset of changes in performance relative to the previous step ($y_t^i$) is updated after each outer step ($D_t = D_{t-1} \cup \{(y_t^i, t, \boldsymbol{h}_t^i)\}_{i=0}^{N-1}$) and used to fit

a Gaussian Process (GP) model. New HPs are then chosen by optimizing an acquisition function that is derived from time-varying GP bandits (Bogunovic et al., 2016) (that are designed for dynamic problems in which the function being optimized changes over time).

PB2-Mix (Parker-Holder et al., 2021) extended the BO approach of PB2 to include categorical HPs. BG-PBT (Wan et al., 2022) further included ordinal HPs while also building upon a BO algorithm (Wan et al., 2021) that is well-suited to mixed-inputs and high-dimensional problems. In BG-PBT, restarts were introduced, with weight information transferred via distillation. We discuss relevant components of BG-PBT when introducing our improvements in Section 3. BG-PBT was strongly focused on enabling Neural Architecture Search (NAS), which we do not address in our work.

Faster Improvement Rate PBT (FIRE-PBT) (Dalibard & Jaderberg, 2021) is a non-BO PBT variant that reduces the greediness of PBT by employing several subpopulations running the PBT algorithm with different objectives targeting short- or long-term performance.

The Multiple-Frequencies PBT (MF-PBT) (Doulazmi et al., 2025) algorithm addresses the issue of setting the step size by running several subpopulations with different step sizes. This performs well with large populations but seems infeasible for the ones that are too small to be split into multiple subpopulations of reasonable size. Doulazmi et al. (2025) further observed significant difference in performance between using their default population size 32 and the smaller 16. In contrast, IPBT is designed to efficiently adjust step size with a single small population, relying on restarts and information reuse.

It should be noted that since PBT variants modify HPs during training, a *schedule* of HPs is optimized, which is known to outperform fixed HP values (Tan & Le, 2021).

### 2.3. Optimization with restarts

Restarting is a powerful idea in optimization: when the optimization procedure stagnates, it is restarted from a different initial point. In repeated local search, the new starting point is chosen randomly, while in *iterated* local search (Lourenço et al., 2003), the new point is chosen so that it is not too far from previously-determined good solutions nor so close as to converge to the same local optimum. A conceptually similar example of such a strategy in deep learning is Stochastic Gradient Descent (SGD) with restarts (Loshchilov & Hutter, 2017), where the learning rate is periodically increased (giving SGD an opportunity to escape from a local optimum) while the weights are retained (so the optimization does not deviate too radically from the previous solution). Furthermore, the idea of restarts has been explored in the field of

evolutionary optimization (Auger & Hansen, 2005).

The weights could also be modified according to the principles of iterated local search. Several algorithms (Sokar et al., 2023; Elsayed et al., 2024) were proposed for modifying previously trained weights to improve downstream optimization. Shrink-perturb (Ash & Adams, 2020) is one such algorithm that was shown to perform well in a variety of contexts (Zaidi et al., 2023; Chebykin et al., 2023): the pretrained weights are modified via shrinking (multiplying by an HP $\lambda_{\text{shrink}}$) and perturbing (adding a random reinitialization of the weights multiplied by an HP $\gamma_{\text{perturb}}$). This both preserves some information present in the weights and introduces randomness that may enable optimization to find a better solution. Alternatively, weight information can be reused by integrating it into a kernel used in Bayesian hyperparameter optimization (Mehta et al., 2025).

## 3. Method

### 3.1. Overview

Each iteration of IPBT follows the general PBT procedure in Figure 2. The time-varying BO algorithm of BG-PBT is used to suggest new HPs. We also follow BG-PBT by starting optimization with $N_{\text{multiplier}}$ times more networks than the population size $N$ and keeping only the best-performing $N$ after the first step.

Efficient adjustment of the step size during an IPBT run is achieved by stopping an iteration once the performance stagnates and restarting with a larger step size. This raises the questions of when to start a new iteration (Section 3.2), how to reuse information from the previous one (Section 3.3), and how to adjust the step size (Section 3.4). The pseudocode of IPBT is provided in Appendix A.

### 3.2. Deciding when to restart

In a black-box setting where nothing is known about the underlying task, one way to select a step size is to empirically try several options. However, fully running a PBT variant several times would reduce efficiency. Efficiency can be preserved by allowing a run with a specific step size to continue only while the performance is strongly improving.

While in BG-PBT the step size was not modified on restarts, the design of its two restart-triggering stopping criteria is relevant. The first one checks that the performance has not improved for $t_{\text{patience}}$ outer steps in a row. This, unfortunately, does not take into account the margin of improvement: a glacially but consistently improving run will not be stopped. This is addressed by the second criterion of BG-PBT: restarting after 40 million environment steps have elapsed since the previous restart (corresponding to 26.6% of the budget in the experiments of Wan et al. (2022); for

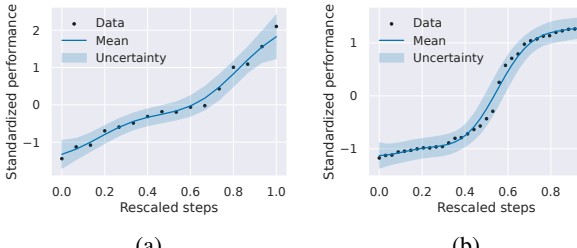

*Figure 3.* Visualizations of the smoothed standardized scores when a trajectory is (a) improving vs. (b) stagnant.

some tasks, this value was set to 60 million steps (40% of the budget)). Since meeting one of the two criteria triggers a restart, the second criterion will eventually terminate the slowly improving run. However, it is not a priori clear how to set the size of the budget that must elapse before a restart is triggered. Additionally, restarting after a fixed budget is inflexible, as a run that improves too slowly cannot be stopped earlier while a strongly improving run cannot be continued for longer.

We propose a data-driven approach that is less susceptible to these issues. The essential questions to be answered are "is the run improving?" and "is it improving too slowly?". To address the first question in the presence of noise, we collect the best performances at each previous outer step, z-score normalize these values, and smooth them via predictive parametric GP regression (Jankowiak et al., 2020) that deals well with heteroscedastic noise[1]. The smoothed value at the current step is compared to that of the previous step, triggering a restart if the current value is not better for $t_{\text{patience}}$ consecutive steps.

Even if the scores are improving, the rate of improvement may be too slow. Determining this is difficult in a black-box setting with no information about what a good score or a good improvement rate is (neither of which can be known for novel tasks). Therefore, the judgment needs to be made based on the observed performance within the run. Our second stopping criterion leverages the standardized performance data, triggering a restart if the performance has not improved by a standard deviation over the last $t_{\text{interval}}$ outer steps. This is task-agnostic and allows the data to speak for itself: if the scores steadily improve, the difference of the current score with that of $t_{\text{interval}}$ steps ago is large (allowing the run to continue while the steady gains persist). If, however, the gains are large at some point but later diminish, the performance $t_{\text{interval}}$ steps ago will eventually be within a standard deviation, since standardization is done based on the entire trajectory (which incorporates information about

---

[1]Model performance includes random variability, which in some settings (like reinforcement learning) can increase as the scores increase during training (i.e., behave heteroscedastically).

the possibility of fast improvements). Figure 3 illustrates both cases.

### 3.3. Reusing information from previous iterations

#### 3.3.1. WEIGHTS

Leveraging information contained in partially-trained weights is crucial for efficiency of the PBT paradigm. Introducing restarts requires figuring out how to transfer this information across them. The two simplest approaches are to either copy the weights from the previous iteration (thus treating the restart as any other outer step of a PBT) or to reinitialize them randomly (as the weights could have caused the stagnation of the previous iteration).

BG-PBT instead relies on distillation to transfer useful behavior from the previous iteration into the next one. This approach allows information transfer even across different architectures but comes with a serious downside: the distillation procedure and its HPs needs to be adapted to the task at hand. Clearly, a fixed distillation procedure cannot be applied across deep learning settings as different as reinforcement learning and image classification. For black-box HPO, a more general approach is desired.

Such an approach should work with the weights directly, since they are the only commonality in all deep learning scenarios. It should both preserve useful information present in the weights and enable downstream optimization to escape from their basin of attraction.

As described in Section 2.3, shrink-perturb (Ash & Adams, 2020) is a straightforward method that consists of copying the weights (multiplied by a constant) and adding noise to them. It is general, operates on weights directly, and both preserves information and injects noise. We integrate shrink-perturb in our algorithm as a method to modify the weights from the previous iteration after a restart. To improve chances of a successful perturbation, before applying shrink-perturb, weights outside the best $\lambda\%$ of the population are replaced with copies of the weights of the best $\lambda\%$.

Additionally, we must consider that weights may end up in unfavorable regions of the optimization space which cannot be escaped via shrink-perturb. To avoid performance degradation in such settings, we randomly reinitialize the weights of half of the population upon restart (instead of shrink-perturbing them).

#### 3.3.2. HYPERPARAMETERS

After restarting, the HPs of the solutions in the new population need to be set. Sampling them randomly neglects the previous experience. On the other hand, relying on past results could also be harmful, since the weights are trained

continuously and HP values that were good initially may not be good for a later stage of the training.

In BG-PBT, an auxiliary global BO procedure was proposed. Firstly, a surrogate $S_i$ is fit on the pairs of (HPs, score) at the latest iteration $i$. Secondly, the best HPs from previous iterations are found and evaluated using $S_i$. The dataset of these HPs and evaluations is used to fit an auxiliary surrogate $S_A$. Finally, $10 \cdot N$ random configurations are sampled, out of which $N$ with the best values of an acquisition function (computed based on $S_A$) are kept. This procedure does not take all the historical trajectories into account (only one from each restart) and uses performance predicted by one model to fit another (which is likely to bias the results).

We propose an alternative approach, specifically, to perform time-varying BO *at the level of restarts*. That is, the HPs at the start of each iteration are optimized to maximize long-term performance within that iteration. This introduces an additional level of dynamic optimization to the PBT setup: while BO optimizes HPs within an iteration, meta BO optimizes the initial HPs across iterations.

Since HP values that lead to good results after a restart are expected to change as the weights are trained, it is reasonable to reuse the time-varying BO procedure of BG-PBT (without any change), fitting it on the long-term performance of initial HPs. To estimate the performance of a set of initial HPs, we track the history of the weights initially trained with them (before potentially switching to other HPs). The best performance achieved by any of these weights is taken as the long-term performance of the initial HPs. Many initial weights are removed from the population relatively quickly, resulting in shorter-term (and worse) performance compared to the best weights. We consider this desirable, as it steers HPO away from the initial HPs that lead to uncompetitive weights.

This procedure aims to learn from past information, which, however, may mislead the algorithm and bias it toward poorly performing local optima. The exploration performed by BO is limited, which is valuable if its model of the problem is correct, but in a black-box setting we must be prepared for any problem, including those that violate the assumptions made by BO. Therefore, similarly to how we randomly reinitialize the weights of half the population, we also sample random HPs for half the population (with sampling independent for weights and HPs).

### 3.4. Adjusting step size

BG-PBT introduced a decreasing step size schedule for certain tasks, which were selected manually because fixed-step-size BG-PBT performed poorly on them. It is not clear how such tasks could be identified in advance. Additionally, a deterministic schedule gives the algorithm no opportunity

to retain a good step size for longer.

Whereas BG-PBT starts with a large step size, we propose starting with a small step size. An intuitive choice is the minimum reasonable amount of weight updates (e.g., an epoch), or 1% of the budget. Starting with a small step size leads to better use of budget in terms of inner steps: as HPs are updated more frequently, fast gains can be made. At the same time, restarts with larger step sizes enable the algorithm to eventually target longer-term performance.

In IPBT, the step size is doubled on each restart. This exponential growth allows larger step sizes to be reached even with limited budget. Alternatively, linear growth can be considered, with the step size increased from $\text{step}_{\text{inner}}$ to $2 \cdot \text{step}_{\text{inner}}$ to $3 \cdot \text{step}_{\text{inner}}$, etc.

## 4. Experiments and Results

### 4.1. Setup

The performance of IPBT is evaluated on the 8 image classification and reinforcement learning tasks that were used by Chebykin et al. (2025) to benchmark PBT variants. The setup of the tasks is followed exactly and the large search spaces are used. The classification tasks entail maximizing accuracy of a ResNet-12 (He et al., 2016) on CIFAR-10/100 (Krizhevsky & Hinton, 2009) and Fashion-MNIST (Xiao et al., 2017) datasets and of a ConvNeXt-T (Liu et al., 2022b) on Tiny ImageNet (Stanford CS231N, 2017). In the reinforcement learning tasks, the score of a proximal policy optimization (Schulman et al., 2017) agent is maximized on the Humanoid, Hopper, Pusher, and Walker tasks from Brax (Freeman et al., 2021). See Appendix B for further details.

Across tasks, the attainable performance scores span different ranges. To enable easy comparison of general algorithm performance, we largely follow the methodology of Agarwal et al. (2021). Specifically, we repeat each experiment 8 times, normalize performance per task across all algorithms, and use stratified bootstrapping to estimate the Interquartile Mean (IQM) and the 95% Confidence Interval (CI). For statistical significance testing, a paired stratified bootstrap test (Efron & Tibshirani, 1993) with Holm correction (Holm, 1979) is used with $\alpha = 0.05$, as described in Appendix C.

We compare IPBT to the 5 PBT variants available in PBT-Zoo (Chebykin et al., 2025): PBT, PB2, PB2-Mix, BG-PBT, and FIRE-PBT (see Section 2.2 and the original publications for details).

Since we are interested in efficient low-budget HPO, the population size of IPBT and PBT variants is set to 8. Although IPBT introduces multiple components with additional HPs, these are designed to work well out of the box and therefore remain fixed across all tasks (see Appendix D for more de-

tails on the HPs). The default values and the components of IPBT are evaluated in our ablation studies (with performance evaluated on 4 tasks: CIFAR-10/100, Humanoid, and Hopper). To ensure unbiased performance estimates, the main experiments were run with seeds different from those used in the ablation studies.

IPBT is run with an initial step size of 1% of the budget (that is automatically adjusted). Other PBT variants are run with the step sizes from (Chebykin et al., 2025): 1%, 3.3%, 10%, 33.3%. For each PBT baseline and for each task, the results across all steps are pooled to estimate untuned performance, while the results corresponding to the step size yielding the maximum performance are considered the tuned performance.

We additionally run non-PBT HPO algorithms: RS as the standard baseline, ASHA (Li et al., 2020) as an efficient multi-fidelity extension of RS, and SMAC3 (Lindauer et al., 2022) as a BO multi-fidelity algorithm. Unlike PBT variants, these algorithms cannot directly search for hyperparameter schedules, placing them at a disadvantage. To address this, we extend their search spaces to include parameters of a cosine learning rate schedule with restarts (Loshchilov & Hutter, 2017), thus allowing them to potentially find learning rate schedules similar to that of IPBT. The baseline algorithms are run with default HPs and the same budget as IPBT and other PBT variants. See Appendix E for further implementation details.

### 4.2. Overall performance

We start by comparing IPBT to previous PBT variants in the one-shot setting, where the step size of each of the PBT variants is not tuned. Figure 4 shows that IPBT strongly outperforms previous PBT variants, clearly demonstrating its utility for dynamic HPO without per-task tuning of the step size (or other HPs of IPBT). The performance of PBT variants relative to each other is reasonable: the most recent Bayesian variant, BG-PBT, achieves the highest IQM, while FIRE-PBT, which uses several subpopulations, performs worst, likely due to the subpopulations being too small.

Selecting the best step size for each task for each PBT variant improves their absolute performance while keeping the relative performance unaffected, as can be seen in Figure 5. Furthermore, selecting the best-performing step size leads to BG-PBT achieving a higher IQM than IPBT, although the difference is not statistically significant. At the same time, IPBT performs statistically significantly better than all other PBT variants, despite using four times less compute (since it was run once, whereas each PBT variant was run four times with different step sizes). We conclude that IPBT should be preferred over other PBT variants, except for BG-PBT, which can achieve better results if its step size is tuned.

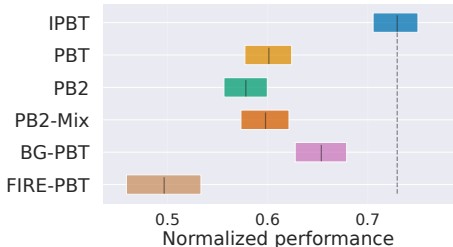

*Figure 4.* Normalized performance across 8 tasks (IQM and CI) of IPBT and *untuned* PBT variants.

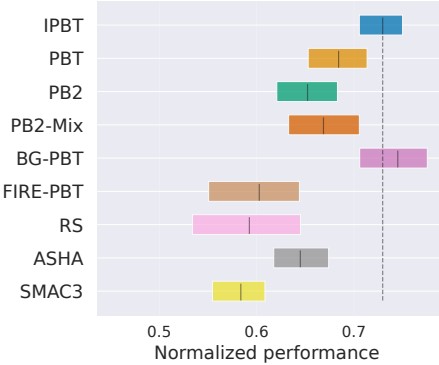

*Figure 5.* Normalized performance across 8 tasks (IQM and CI) of IPBT, *tuned* PBT variants, and baselines.

The need for tuning step sizes of PBT baselines can be clearly seen in Figure 6, which shows that PBT variants achieve the best performance with larger step sizes (10%, 33.3%) on Humanoid, and smaller (1%, 3.3%) on Hopper.

We further compare IPBT with non-PBT HPO algorithms that can be run out-of-the-box with the same small computation budget. Figure 5 shows that IPBT achieves a much better IQM (statistically significant) than RS, ASHA, and SMAC3. The unexpectedly poor performance of SMAC3 (on par with RS) led us to investigate, revealing that it is likely due to the algorithm allocating a large fraction of the budget to cheap evaluations and running few expensive ones. This strategy works well for some tasks (e.g., Hopper in Figure 6) but not others (e.g., Humanoid in Figure 6). While HPs of SMAC3 (and ASHA) could potentially be adjusted per-task to achieve better results, this would increase the computational budget and serve as a disadvantage compared to IPBT that performs well without per-task HP adjustment.

### 4.3. Ablation studies

In this section, we perform ablation studies by varying components of IPBT. The results are shown in Figure 7 and will be discussed in the order presented there.

Firstly, replacing our data-driven mechanism of stagnation detection with the procedure used in BG-PBT reduces IQM, demonstrating the benefit of a more flexible mechanism.

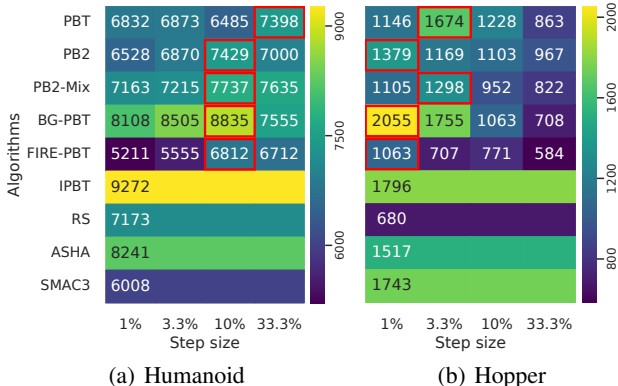

*Figure 6.* IQM of scores of PBT variants with different step sizes (the best score of each variant is framed in red), IPBT, and baselines on Humanoid and Hopper (see Appendix F for the other tasks).

We then investigate components related to the reuse of weight information. Since shrink-perturb is a middle ground between copying weights exactly (which corresponds to parameters $(1.0, 0.0)$) and reinitializing them randomly (which corresponds to $(0.0, 1.0)$), we try both of these options and find that they worsen performance substantially. Next, we vary $\lambda_{\text{shrink}}$ from its default of $0.2$ to $0.4$ and $0.1$, observing a reduction in the IQM that highlights the importance of this HP. Another component related to weight reuse is random reinitialization of weights of half the models on a restart. Disabling it moderately reduces performance, highlighting the importance of injecting randomness to escape local minima. In Appendix G, we additionally replace shrink-perturb with the distillation procedure of BG-PBT on reinforcement learning tasks, and find performance to be similar (which means that performance is not sacrificed when the task-agnostic shrink-perturb is used).

Investigating the HP information reuse, we find that replacing our HP reinitialization strategy based on time-varying BO with either BG-PBT's global BO strategy or random sampling lowers the IQM. However, if no random sampling is done at all, performance decreases, albeit marginally. We conclude that some noise injection improves overall performance.

In IPBT, step size is increased exponentially (doubled) on each restart. Increasing it linearly is worse, while keeping it constant is the worst by a substantial margin. This shows that adjusting the step size is essential to the success of IPBT.

Finally, we investigated whether restarting with twice the population size (and retaining the best half after the first step) is beneficial compared to not doing so ("Population multiple 1") or starting with thrice the size. Our results indicate that the default value of 2 strikes a good balance between discarding models with poor initial performance

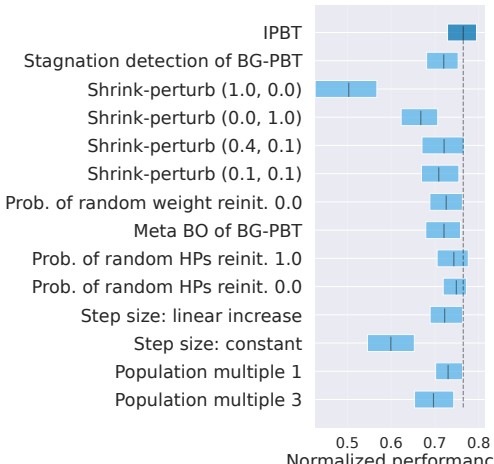

*Figure 7.* Ablation studies of IPBT: normalized performance (IQM and CI) across 4 tasks.

and avoiding excessive bias toward models with good performance after a single outer step.

### 4.4. Qualitatively inspecting a discovered schedule

IPBT is designed to naturally discover schedules with restarts. It can find such schedules not only for the learning rate (where restarts are well-established) but also other hyperparameters, as can be seen in Figure 8. For example, the number of Rand-Augment[2] operations is reset to 1 on each restart (and then increases within the iteration), while the initial magnitude of the augmentations increases across restarts (and appears to first increase and then decrease during the iteration). This serves as a clear example of how within- and across-iteration optimization both contribute to dynamically optimizing HPs. Quantitative evaluation of training from scratch with discovered schedules is provided in Appendix H.

## 5. Discussion and Conclusion

In this paper, we have introduced IPBT, a powerful HPO algorithm that demonstrates superior performance across 8 image classification and reinforcement learning tasks. By automatically adjusting its step size, IPBT substantially outperforms previous PBT variants with untuned step sizes and matches or exceeds their performance when their step sizes are tuned (while avoiding the cost of tuning). Moreover, IPBT significantly surpasses other popular HPO algorithms, such as RS, ASHA, and SMAC3.

IPBT achieves good results thanks to its iterative nature.

---

[2]RandAugment (Cubuk et al., 2020) is an image augmentation procedure that applies several random augmentations of a specific magnitude (e.g., for rotation, different magnitudes correspond to different maximum angles by which an image can be rotated).

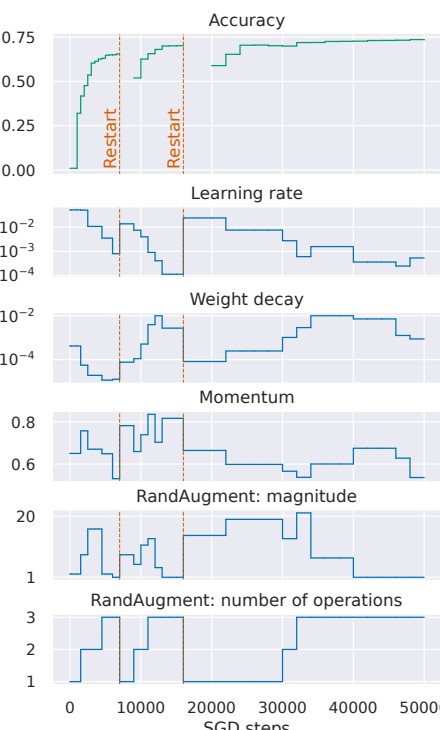

*Figure 8.* Accuracy and HP schedule of the best solution from an IPBT run on CIFAR-100.

Restarts allow escape from local optima in both weight and HP space, while efficiency is retained through information reuse across restarts: weights are adapted via shrink-perturb, and the initial HPs of each restart are sampled using a time-varying BO procedure. The decision when (and whether) to trigger a restart is made by our data-driven mechanism, allowing for flexibility based on performance dynamics.

Our ablation studies empirically demonstrated the benefit of the components of IPBT. We have generally observed that injecting some noise both into the weights and into HPs is beneficial in comparison to injecting no noise or relying on random exploration entirely. The component that seems most interesting for further investigation in future work is shrink-perturb (the mechanism for weight reuse), both in terms of how exactly its parameters influence HPO, and whether it can be productively replaced by a different method for partially resetting the weights.

While good performance of IPBT without per-task tuning of its hyperparameters is encouraging, investigation of potential benefits of such tuning could be an interesting avenue of future work.

Other promising directions for future work (and current limitations) include adaptation to multi-objective optimization (Dushatskiy et al., 2023) and NAS (Franke et al., 2021), as well as evaluation with different architectures, such as

Transformer (Vaswani et al., 2017).

We designed IPBT to work well for black-box problems and small computation budgets (equivalent to 8 training runs). Algorithms that incorporate prior information about the problem are likely to outperform IPBT if such information is available (Mallik et al., 2023). Similarly, larger computational budgets are likely to enable state-of-the-art BO algorithms (Cowen-Rivers et al., 2022; Yu et al., 2025) to achieve better results.

Nonetheless, enabling efficient search for HP schedules within the wall-clock time of a single training run remains a notable advantage of PBT variants in general and IPBT in particular. Strong out-of-the-box performance of IPBT further marks it as a PBT variant of choice for efficiently optimizing HPs of deep learning tasks in low-budget settings.

## Acknowledgments

This work is part of the research project DAEDALUS which is funded via the Open Technology Programme of the Dutch Research Council (NWO), project number 18373; part of the funding is provided by Elekta and ORTEC LogiqCare.

## Impact Statement

This paper presents work whose goal is to advance the field of Machine Learning. There are many potential societal consequences of our work, none of which we feel must be specifically highlighted here.

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

# Appendix

## A. Pseudocode of IPBT

The pseudocode of IPBT is listed in Algorithms 1, 2, 3; see the main text and the source code for further details.

---

**Algorithm 1** IPBT

---

**Input:** population size $N$, maximum number of inner steps $\text{step}_{\max}$, step size $\text{step}_{\text{inner}}$, selection parameter $\lambda$, hyperparameter restart parameters $t_{\text{patience}}, t_{\text{interval}}$, weight restart parameters $\lambda_{\text{shrink}}, \gamma_{\text{perturb}}$

1: $\text{pop} \leftarrow \{$a pair of random HPs $\boldsymbol{h}^i$ and weights $\theta^i\}_{i=0}^{N-1}$
2: $\text{D} \leftarrow \{\}$ # Dataset of performance differences
3: $\text{D}_{\text{it}} \leftarrow \{\}$ # Dataset of performance differences within the current iteration (i.e., before a restart)
4: $\text{step} \leftarrow 0$
5: **while** $\text{step} < \text{step}_{\max}$ **do**
6: $\quad \text{step} \leftarrow \text{step} + \text{step}_{\text{inner}}$
7: $\quad$ **for** $i \leftarrow 0$ **to** $N - 1$ **do** # in parallel
8: $\qquad$ Train $\theta^i$ for $\text{step}_{\text{inner}}$ steps using $\boldsymbol{h}^i$
9: $\qquad \text{perf}_{\text{i}}^{\text{step}} \leftarrow \texttt{evaluate}(\theta^i)$
10: $\qquad \text{info}_{\text{i}} \leftarrow \left(\text{perf}_{\text{i}}^{\text{step}} - \text{perf}_{\text{i}}^{\text{step}-\text{step}_{\text{inner}}}, \text{step}, \boldsymbol{h}^i\right)$
$\qquad$ # store difference to the previous performance and the HPs
11: $\qquad \text{D}_{\text{it}} \leftarrow \text{D}_{\text{it}} \cup \{\text{info}_{\text{i}}\}$
12: $\qquad \text{D} \leftarrow \text{D} \cup \{\text{info}_{\text{i}}\}$
13: $\quad \text{restart} \leftarrow \texttt{check\_restart}(\text{D}_{\text{it}}, t_{\text{patience}}, t_{\text{interval}})$
$\quad$ # § 3.2
14: $\quad$ Sort pop by perf
15: $\quad$ **if not** restart **then**
16: $\qquad$ Replace the worst $\lambda\%$ with the copies of the best $\lambda\%$ (selected randomly)
17: $\qquad$ Fit a surrogate on $\text{D}_{\text{it}}$, use BO to set HPs of the copies
18: $\quad$ **else**
19: $\qquad$ Replace everything but the best $\lambda\%$ with the copies of the best $\lambda\%$ (selected randomly)
20: $\qquad \text{pop} \leftarrow \texttt{reinit}(\text{pop}, \text{D}, \lambda_{\text{shrink}}, \gamma_{\text{perturb}})$
$\qquad$ # § 3.3
21: $\qquad \text{step}_{\text{inner}} \leftarrow 2 \cdot \text{step}_{\text{inner}}$ # § 3.4
22: $\qquad \text{D}_{\text{it}} \leftarrow \{\}$

---

**Algorithm 2** `check_restart`

---

**Input:** current-iteration dataset $\text{D}_{\text{it}}$, restart parameters $t_{\text{patience}}, t_{\text{interval}}$

1: $\text{p} \leftarrow$ best performances at previous steps in the current iteration
2: $\text{z} \leftarrow$ z-score normalized $\text{p}$
3: $\text{s} \leftarrow$ z smoothed via GP regression
4: $m \leftarrow |\text{z}|$
5: $\text{stalled} \leftarrow \bigwedge_{k=m-t_{\text{patience}}+1}^{m} s_k \leq s_{k-1}$ # no improvement for $t_{\text{patience}}$ consecutive steps
6: $\text{slow} \leftarrow (z_m - z_{m-t_{\text{interval}}}) < 1$ # improvement over the last $t_{\text{interval}}$ steps is below a standard deviation
7: **return** stalled **or** slow

---

**Algorithm 3** `reinit`

---

**Input:** population pop, dataset D, shrink-perturb parameters $\lambda_{\text{shrink}}, \gamma_{\text{perturb}}$

1: $\text{R}_\theta \leftarrow$ uniformly sample $\text{round}(N/2)$ indices from $\{0, \dots, N-1\}$ without replacement
2: **for** $i \leftarrow 0$ **to** $N - 1$ **do**
3: $\quad$ **if** $i \in \text{R}_\theta$ **then**
4: $\qquad \theta^i \leftarrow$ random weight initialization
5: $\quad$ **else**
6: $\qquad \theta_{\text{rand}}^i \leftarrow$ random weight initialization
7: $\qquad \theta^i \leftarrow \lambda_{\text{shrink}}\theta^i + \gamma_{\text{perturb}}\theta_{\text{rand}}^i$ # shrink-perturb
8: $\text{R}_{\text{h}} \leftarrow$ uniformly sample $\text{round}(N/2)$ indices from $\{0, \dots, N-1\}$ without replacement
9: Fit time-varying meta BO on D # initial HPs and descendants' best performances
10: **for** $i \leftarrow 0$ **to** $N - 1$ **do**
11: $\quad$ **if** $i \in \text{R}_{\text{h}}$ **then**
12: $\qquad \boldsymbol{h}^i \leftarrow$ sample randomly
13: $\quad$ **else**
14: $\qquad \boldsymbol{h}^i \leftarrow$ sample via meta BO
15: **return** pop

---

## B. Tasks, search spaces, and evaluation

Our experiments are run on the tasks previously used for evaluating PBT variants (Chebykin et al., 2025). In image classification tasks, hyperparameters of a ResNet-12 (He et al., 2016) are optimized on CIFAR-10/100 (Krizhevsky & Hinton, 2009), and Fashion-MNIST (Xiao et al., 2017) datasets for 50,000 Stochastic Gradient Descent (SGD) steps; and hyperparameters of a ConvNeXt-T (Liu et al., 2022b) are optimized on Tiny ImageNet (Stanford CS231N,

2017) for 150,000 steps. In reinforcement learning tasks, hyperparameters of a proximal policy optimization (Schulman et al., 2017) agent are optimized on tasks implemented in Brax (Freeman et al., 2021) (Humanoid, Hopper, Pusher, Walker) during 150,000,000 environment interactions.

For PBT variants, the *large* search spaces of (Chebykin et al., 2025) are used for both classification (Table 1) and reinforcement learning (Table 2) tasks. For the non-PBT baselines, the search spaces are extended with hyperparameters of a cosine learning rate schedule with restarts (Table 3). Since the non-PBT baselines are run with an iteration equal to 1% of the budget, the number of iterations before the first restart is between 5% and 50% of the budget.

We did not change evaluation metrics, using accuracy for image classification, and task score for reinforcement learning

*Table 1.* Search space for the classification tasks (the *large* search space from (Chebykin et al., 2025)).

| Hyperparameter | Type | Exponent base | Range |
|---|---|---|---|
| Learning rate | real | 10 | [-6, 0] |
| Number of augmentations | integer | ✗ | [1, 4] |
| Augmentation strength | integer | ✗ | [1, 30] |
| Weight decay | real | 10 | [-8, -2] |
| Momentum | real | ✗ | [0.5, 0.999] |

*Table 2.* Search space for the reinforcement learning tasks (the *large* search space from (Chebykin et al., 2025)).

| Hyperparameter | Type | Exponent base | Range |
|---|---|---|---|
| Learning rate | real | 10 | [-4, -3] |
| Entropy coefficient | real | 10 | [-6, -1] |
| Batch size | integer | 2 | [8, 10] |
| Discount factor | real | ✗ | [0.9, 0.9999] |
| Unroll length | integer | ✗ | [5, 15] |
| Reward scaling | real | ✗ | [0.05, 20] |
| Number of updates per epoch | integer | ✗ | [2, 16] |
| GAE parameter | real | ✗ | [0.9, 1] |
| Clipping parameter | real | ✗ | [0.1, 0.4] |

*Table 3.* Additional hyperparameters of a cosine learning rate schedule with restarts used by non-PBT baselines.

| Hyperparameter | Type | Exponent base | Range |
|---|---|---|---|
| Number of iterations before the first restart | integer | ✗ | [5, 50] |
| Multiplier for the number of iterations between restarts | integer | ✗ | [1, 2] |
| Minimum learning rate | real | 10 | [-8, -6] |

tasks.

All performance claims in the paper are based on the test performances of the best models of each run of an algorithm. For algorithms with restarts (IPBT, BG-PBT, non-PBT baselines), the best model is chosen based on validation performance achieved either before a restart or at the end of training, while for the PBT variants without restarts, the best model is chosen based on the validation performance at the end of training.

## C. Statistical testing

In the main text, in order to compare algorithms while taking performance of all tasks into account, the IQM of the normalized performance across tasks and seeds is estimated via stratified bootstrapping (Agarwal et al., 2021). To assess statistical significance of differences between the IQMs of two algorithms, a paired stratified bootstrap test (Efron & Tibshirani, 1993) is executed with 50,000 replicates.

In every replicate, we sample seeds with replacement, use those seeds as indices for both algorithms across all tasks (thus getting a paired sample), and compute the difference

*Table 4.* P-values (rounded to 5 decimal places) of pair-wise statistical tests of IPBT against baselines (sorted by p-value)

| Algorithm | Rejected $H_0$ | p | p (Holm) |
|---|---|---|---|
| FIRE-PBT | Yes | 0.00002 | 0.00016 |
| SMAC3 | Yes | 0.00002 | 0.00016 |
| Random search | Yes | 0.00004 | 0.00024 |
| ASHA | Yes | 0.00012 | 0.00060 |
| PB2 | Yes | 0.00022 | 0.00088 |
| PB2-Mix | Yes | 0.00810 | 0.02430 |
| PBT | Yes | 0.02070 | 0.04140 |
| BG-PBT | No | 0.49701 | 0.49701 |

between the IQMs. The replicate differences are then centered by the observed IQM, and a two-sided p-value is obtained from the (corrected) proportion of centered replicates whose magnitude exceeds the observed difference (Davison & Hinkley, 1997). The code for this procedure is included in our source code release.

IPBT is pairwise compared with 5 tuned PBT variants and 3 baselines (RS, ASHA, SMAC3) — 8 tests in total. The target significance level is 0.05, and the Holm correc-

tion (Holm, 1979) is used to control the family-wise error rate. The results in Table 4 show that performance of IPBT is significantly different from that of all algorithms except for BG-PBT.

## D. Hyperparameters

The hyperparameters of all the algorithms are listed in the configuration files included in the source code release. In this section, we discuss the setting of hyperparameters of novel components of IPBT.

**Shrink-perturb**   The parameter values used in the main experiments are $(0.2, 0.1)$. The results when using $(0.4, 0.1)$ and $(0.1, 0.1)$ are reported in the ablation study in the main text. Additionally, the setting $(0.5, 0.5)$ was tried during early development (on the CIFAR-10 task) and found to perform worse.

**Stagnation detection**   The parameters for restart detection used in the experiments were set to $t_{\text{patience}} = 3$ and $t_{\text{interval}} = 15$. These values were chosen based on visual inspection of performance traces during preliminary experiments. While $t_{\text{interval}}$ was not varied, values of 2, 3, and 4 were tested for $t_{\text{patience}}$ using an early version of the algorithm on CIFAR-10/100 and Brax Humanoid tasks. The setting $t_{\text{patience}} = 3$ yielded the highest IQM.

**Initial step size**   In the experiments, the initial step size is set to 1% of the budget. We additionally experiment with settings of 0.5% and 3%, results are shown in an ablation study in Figure 9. It can be observed that reducing the step size to 0.5% leads to only a slight decrease in IQM, as IPBT can increase the step size when needed. Starting with a larger step size of 3% reduces performance more, as IPBT is not designed to decrease the step size and larger steps lead to faster budget exhaustion.

**Strategy of step size increase**   The step size is doubled on each restart. We additionally tried increasing it linearly, reporting the results in the ablation study in the main text.

All preliminary experiments were run with seeds distinct from those used for the ablation studies and the main experiments.

## E. Implementation details

**Hardware and software**   The configuration of the servers used to run experiments: 3 Nvidia A5000 GPUs, 2 Intel(R) Xeon(R) Bronze 3206R CPUs, and 96 GB of RAM. The OS is Fedora Linux 36. The random seeds and the versions of all used frameworks and libraries are specified in the configuration files included in the source code release.

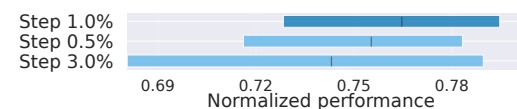

*Figure 9.* Extra ablation study on the initial step size of IPBT: normalized performance (IQM and CI) across 4 tasks (same setup as the ablation study in the main text).

**Computational cost of a restart**   Upon a restart, new hyperparameters are produced via a meta-BO procedure while the weights are adapted via shrink-perturb. In our experiments, these steps took $\approx 80$ CPU-seconds and $\approx 0.5$ seconds correspondingly. Since restarts occur only a few times per run, their cost is negligible, especially as larger models are trained on larger datasets (e.g., in our TinyImageNet experiments, training for 1% of the budget takes $\approx 1300$ seconds on an A5000 GPU).

**SMAC3**   Directly using the latest version of SMAC3 (2.3.1) and setting the total budget in line with the documentation caused the algorithm to stop after only $\approx 75\%$ of the budget was used. We believe this issue stems from a discrepancy between the theoretically calculated budget usage, and the practical asynchronous implementation. To address this, we forked the code and introduced an alternative stopping criterion that monitors the actual budget consumed. This resulted in nearly full budget usage and improved performance on CIFAR-10 in preliminary experiments.

**BG-PBT**   As mentioned in Section 3.3, Wan et al. (2022) use either 26.6% or 40% of the budget in their second stopping criterion (depending on the task). For uniformity, we use 30% for all tasks. We enable this criterion only with the step size of 1% (that is the closest to the original setting of BG-PBT) because otherwise too few steps are done before a forced restart is triggered. We also enable distillation only in this setting. Additionally, when the step size is 33.3%, we change $N_{\text{multiplier}}$ from the default 3 to 1 because otherwise the entire budget is spent on random initialization.

## F. Results for individual tasks

The performance of IPBT and the baselines on individual tasks not included in Figure 6 is shown in Figure 10. IPBT achieves consistently good (though not always the best) results without any change to its hyperparameters.

## G. Replacing shrink-perturb with distillation

In BG-PBT, a distillation procedure is used to transfer information stored in weights across restarts. We replace shrink-perturb in IPBT with this procedure, and evaluate it on the Humanoid and Hopper tasks (as the distillation is

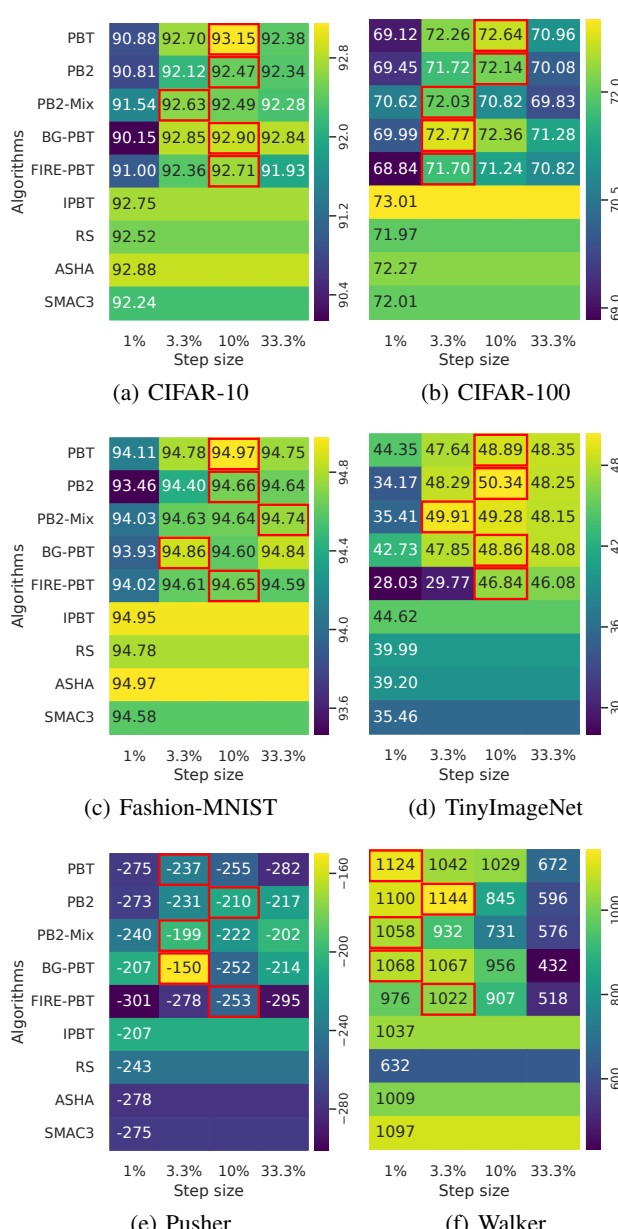

*Figure 10.* IQM of scores of PBT variants with different step sizes (the best score of each variant is framed in red), IPBT, and baselines.

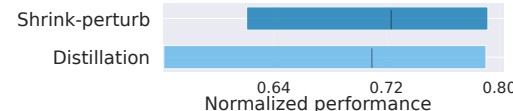

*Figure 11.* Extra ablation study on the weight reuse mechanism of IPBT: normalized performance (IQM and CI) across the Humanoid and Hopper tasks.

*Table 5.* IQM and IQR of accuracies achieved by IPBT and by using the best schedules discovered by IPBT to train new weights from scratch.

| Dataset | IPBT | Replay |
|---|---|---|
| CIFAR-100 | $73.0_{(72.6,73.3)}$ | $73.4_{(73.1,73.9)}$ |
| CIFAR-10 | $92.8_{(92.6,92.9)}$ | $93.0_{(92.9,93.2)}$ |
| Fashion-MNIST | $95.0_{(94.8,95.1)}$ | $95.0_{(94.9,95.1)}$ |

ules can be used to train networks with different weight initializations from scratch.

To answer this question, we start by conducting additional experiments on the CIFAR-10/100, and Fashion-MNIST datasets. For each of the 8 runs of IPBT on each dataset, we extract the best hyperparameter schedule produced by IPBT and use it to train models with different initializations and using different random seeds.

In Table 5, we report IQM and IQR of accuracy on test data. It can be seen that on these classification tasks, the performance of a replay is as good as that of the original optimization run. Thus, the schedules discovered by IPBT can be used to train new models cheaply, with a single model rather than a population[3].

We further investigate the effect of schedule replay in the reinforcement learning setting, where the training is generally less stable and thus may benefit from the stability and robustness of training a population (rather than a single network). Note that picking the best-performing schedule found by a non-population-based algorithm equivalently increases performance and stability when more than one network is trained.

Consequently, we repeat the schedule replay experiment on the Humanoid task using the schedules discovered by IPBT and ASHA (the best non-PBT baseline on this task), reporting the results in Table 6.

It can be seen that both algorithms show worse performance and stability, which is in line with the inherent randomness and instability of reinforcement learning that is not fully

specific to reinforcement learning tasks). Figure 11 shows that the performance is approximately the same, meaning that task-agnostic shrink-perturb matches task-specific distillation.

## H. Training with discovered schedules

IPBT and other PBT variants continuously train weights while dynamically adjusting hyperparameters. This is efficient but raises the question of whether the discovered sched-

---

[3]Note that shrink-perturb is applied on restarts during replay but it does not require a population and takes less than a second, thus being effectively free — in contrast to, e.g., knowledge distillation used by BG-PBT, the runner-up PBT variant.

*Table 6.* IQM and IQR of scores on the Humanoid task achieved by IPBT and ASHA, and by using the best discovered schedules to train new weights from scratch.

| Method | Original | Replay |
|--------|----------|--------|
| IPBT | $9272_{(8831,9737)}$ | $6939_{(4372,8887)}$ |
| ASHA | $8241_{(7241,8941)}$ | $6197_{(4449,7089)}$ |

addressed by using an optimized schedule. Nonetheless, IPBT achieves higher IQM than ASHA not only in the original experiments but also when the optimized schedules are replayed, demonstrating that the improvement of IPBT over ASHA persists.

Our conclusion is that replaying the optimal schedule can be expected to give similar results in more stable settings such as image classification but diminished results in less stable settings such as reinforcement learning. In such settings, it may be better to treat PBT variants as weight training methods rather than hyperparameter optimization methods, as is sometimes done in the literature (Liu et al., 2022a).

