# OpenReview forum: "Iterated Population Based Training with Task-Agnostic Restarts"
_ICML.cc/2026/Conference — ICML 2026 regular_

### Official Review · Reviewer_rH8h · 2026-03-10

**Soundness:** 3
**Presentation:** 3
**Significance:** 3
**Originality:** 3
**Overall Recommendation:** 4
**Confidence:** 4

**Summary:**

This paper introduces Iterated Population Based Training (IPBT), a new variant of the Population Based Training (PBT) algorithm designed to address the challenges of setting the "step size"—the number of inner training steps performed before hyperparameter (HP) updates occur.
The algorithm operates through the following key mechanisms:1.Task-Agnostic Restarts；2.Data-Driven Stagnation Detection；3.Task-Agnostic Information Reuse.
The main contribution of this paper is the introduction of Iterated Population Based Training (IPBT), an algorithm that addresses the critical challenge of manually tuning the step size in standard PBT by automatically adjusting this meta-hyperparameter through a system of task-agnostic, data-driven restarts. By incorporating mechanisms such as automated stagnation detection, weight reuse via a "shrink-perturb" technique, and time-varying meta-Bayesian optimization, IPBT eliminates the need for user-defined heuristics and achieves competitive or superior performance across a diverse set of image classification and reinforcement learning tasks compared to existing state-of-the-art PBT variants and baselines.

**Compliance With Llm Reviewing Policy:**

Affirmed.

**Key Questions For Authors:**

1. Are there any task-specific hyperparameters within the IPBT framework itself (e.g., threshold sensitivity for stagnation detection or restart criteria) that require manual tuning?

2. Could you provide a quantitative comparison of the "management overhead" (the compute cost of the meta-BO and weight manipulation) relative to the actual neural network training cost?

**Limitations:**

See Weaknesses.

**Strengths And Weaknesses:**

Strengths

1.The paper provides a rigorous empirical evaluation across 8 diverse tasks in two distinct domains: reinforcement learning (Brax) and image classification (CIFAR, Tiny ImageNet). The authors ensure fairness by maintaining a fixed budget across all methods (8 full training runs) and using standardized metrics (IQM and 95% CI) with bootstrapping for statistical significance.

2.The submission is well-structured and the narrative is clear. It effectively contextualizes the problem by identifying "step size" as a critical yet overlooked meta-hyperparameter in PBT literature. Figure 1 provides a helpful conceptual overview of the IPBT framework and its iterative nature.

3.The work addresses a high-impact practical problem: the sensitivity of PBT to the update interval, which often requires manual tuning. By automating this process in a task-agnostic manner, IPBT lowers the barrier for practitioners to use dynamic HPO in resource-constrained settings.

4.The novelty stems from the creative combination of an iterative restart mechanism with automatic step-size doubling. The introduction of a data-driven, task-agnostic stagnation detection criterion (using GP regression to normalize performance) is a notable improvement over the fixed-budget or patience-based restarts used in prior work like BG-PBT.


Weaknesses:

1.Dependent on fixed hyperparameters for its novel components.While IPBT eliminates per-task step size tuning, its core components (e.g., shrink-perturb parameters (0.2,0.1), stagnation detection thresholds \mathrm{t}_{\mathrm{patience}}\mathrm{=3}, \mathrm{t}_{\mathrm{interval}}\mathrm{=15}) use fixed values across all tasks. These values are determined via preliminary experiments on a small subset of tasks, and there is no exploration of their adaptability to more diverse or niche tasks (e.g., NLP, graph learning).

2.Limited performance on large-budget HPO scenarios.IPBT is explicitly optimized for low-budget HPO (8 full training runs). The paper acknowledges that state-of-the-art BO algorithms with larger computational budgets will likely outperform IPBT, limiting its utility for settings where extensive compute is available.

---

> ### Author Rebuttal · Authors · 2026-03-30
>
> We thank the reviewer for the helpful feedback.
>
> > While IPBT eliminates per-task step size tuning, its core components \[...\] use fixed values across all tasks. These values are determined via preliminary experiments on a small subset of tasks, and there is no exploration of their adaptability to more diverse or niche tasks (e.g., NLP, graph learning).
>
> The reviewer is correct that we have not explored the adaptability of fixed values to tasks beyond the 8 image classification and reinforcement learning tasks we considered. This remains interesting future work, which we will mention in Section 5 of the updated paper.
>
> > [Key question 1:] Are there any task-specific hyperparameters within the IPBT framework itself (e.g., threshold sensitivity for stagnation detection or restart criteria) that require manual tuning?
>
> IPBT showed good empirical peformance with fixed hyperparameters across all tasks (including those not seen when setting hyperparameter values), as described in Section 4. Therefore, manual per-task tuning is not required (although it potentially could further boost performance).
>
> > [Key question 2:] Could you provide a quantitative comparison of the "management overhead" (the compute cost of the meta-BO and weight manipulation) relative to the actual neural network training cost?
>
> In our experiments, the meta-BO procedure took ~80 CPU-seconds to produce new solutions upon a restart (and weight adaptation via shrink perturb took ~0.5 seconds). As both of these happen approximately 3-4 times per run, the cost is negligible, especially as larger models are trained on larger datasets.
>
> For example, in our TinyImageNet experiments, training for 1% of the budget (corresponding to the default step size of IPBT) takes ~1300 seconds on an A5000 GPU, which is much longer than the total time spent on meta-BO and shrink-perturb.
>
> We will update the manuscript with this information.

---

> > ### Author Rebuttal · Reviewer_rH8h · 2026-04-04
> >
> > My concerns are addressed and I hope the authors revise the paper as they promised. I will engage in the Reviewer-AC phase to champion this paper.

---

### Official Review · Reviewer_jcDR · 2026-03-12

**Soundness:** 1
**Presentation:** 1
**Significance:** 1
**Originality:** 1
**Overall Recommendation:** 1
**Confidence:** 5

**Summary:**

This study proposes an iterated population-based training (IPBT) for hyperparameter optimization. The IPBT consists of three main components, including determination of restart, weights inheritance, and adaptive step size. Experiments examine the performance in comparison to five PBT variants on image classification and reinforcement learning problem.

**Compliance With Llm Reviewing Policy:**

Affirmed.

**Final Justification:**

As the main concerns mentioned in my review comments were only partially resolved that my concerns in soundness, presentation, significance, and originality were not well-addressed in the rebuttal. I kept my rating "1: Strong Reject".

**Key Questions For Authors:**

1. The maximum training steps are much larger than those used in image classification and reinforcement learning tasks. How is the performance if similar computational budgets are used?
2. How is the performance of IPBT if initial step size increases?

**Limitations:**

The computational cost may be a critical concern for practical use.

**Strengths And Weaknesses:**

Strengths
- This study combines existing techniques with different implementation for improving the performance.

Weaknesses
1. The design of experiments should be improved. The comparisons are under a simple architecture. The effectiveness should be validated on more advanced methods.
2. The presentation should be further enhanced. The use of definition, equation, and algorithm may improve the clarity.
3. The novelty should be further clarified. All the three main components are common techniques in metaheuristics. The consideration of dynamically adjust restart timing, information perturbation/abandon, and increase/decrease step size and their combinations are well-known. For example, the famous covariance matrix adaptation evolution strategy (CMA-ES) has considered all these ideas.
4. The significance should be elaborated. The IPBT has similar performance in comparison to BG-PBT on image classification (Fig. 5) and reinforcement learning problem (Fig. 6).

---

> ### Author Rebuttal · Authors · 2026-03-30
>
> We thank the reviewer for the helpful feedback.
>
> > The design of experiments should be improved. The comparisons are under a simple architecture. The effectiveness should be validated on more advanced methods.
>
> We would like to point out that for the Tiny ImageNet experiments, the ConvNeXt-T architecture is used, which is a modern convolutional architecture. Furthermore, since our key contribution is a hyperparameter optimization algorithm, our comparisons include state-of-the-art baselines, both in the PBT family (BG-PBT) and outside it (SMAC3).
>
> > The presentation should be further enhanced. The use of definition, equation, and algorithm may improve the clarity.
>
> In addition to our description of the IPBT algorithm in Section 3, its pseudocode is provided in Appendix A. We will further extend it with the pseudocode of the algorithms' components (which are currently only described textually in Section 3).
>
> > The novelty should be further clarified. All the three main components are common techniques in metaheuristics. The consideration of dynamically adjust restart timing, information perturbation/abandon, and increase/decrease step size and their combinations are well-known. For example, the famous covariance matrix adaptation evolution strategy (CMA-ES) has considered all these ideas.
>
> We agree that the broad ideas behind the components of our algorithm have been previously studied by the metaheuristics community, with our manuscript including citations to that effect (which we will extend with the CMA-ES citation).
>
> The novelty of our work lies in coming up with concrete realizations of these general ideas that jointly allow us to address the key weakness of the PBT family of algorithms (sensitivity to the step size). This enables IPBT to effectively optimize deep learning hyperparameters, which is an important problem for the field. Our deep-learning-specific adaptations are vital, since efficiency requires model weights to be properly reused, which is not the focus of meta-heuristics work (with the potential exception of neuroevolution).
>
> > The significance should be elaborated. The IPBT has similar performance in comparison to BG-PBT on image classification (Fig. 5) and reinforcement learning problem (Fig. 6).
>
> The key benefit of IPBT is working out-of-the-box for diverse tasks on a small budget. It can be seen in Figure 4 that when baseline algorithms (including BG-PBT) are not tuned, IPBT strongly outperforms all of them. When the baselines are tuned, IPBT still outperforms all of them, except for BG-PBT, with which it is tied, as the reviewer correctly observes in Fig. 5. However, BG-PBT requires four times more computational effort than IPBT to achieve these results, marking IPBT as a superior choice for efficient out-of-the-box hyperparameter optimization.
>
> > [Key question 1:] The maximum training steps are much larger than those used in image classification and reinforcement learning tasks. How is the performance if similar computational budgets are used?
>
> We would like to clarify that the maximum number of training steps is kept constant across all evaluated algorithms. The restarts performed by IPBT are done within a single run and do not affect the number of training steps, making the comparisons fair. We will state this point more clearly in the updated paper.
>
> > [Key question 2:] How is the performance of IPBT if initial step size increases?
>
> We have varied the initial step size in an additional ablation study in Appendix D. It can be seen in Fig. 9 that increasing the initial step size slightly reduces performance, as IPBT is designed to start with a small step size (for efficiency) and increase it as optimization progresses.

---

> > ### Author Rebuttal · Reviewer_jcDR · 2026-04-01
> >
> > Thank you for responding my comments. However, my major concerns still remain with additional questions.
> > 1. How is the performance improvement on state-of-the-art methods in reinforcement learning and image classification?
> > 2. How is the performance when using a common computational budget in reinforcement learning such as a million environmental steps?
> > 3. Are there any studies using restart and re-using weight information for hyperparameter tuning? If so, discussion with proper citations should be added.
> > 4. Where are the evidences that BG-PBT requires more running time?

---

> > > ### Author Response · Authors · 2026-04-08
> > >
> > > > How is the performance improvement on state-of-the-art methods in reinforcement learning and image classification?
> > >
> > > The key contribution of our paper is introducing a hyperparameter optimization algorithm that outperforms or matches prior algorithms without per-task tuning. To be able to confidently claim this, we have intentionally selected to optimize relatively small (yet realistic and modern) models that are not prohibitively computationally expensive. This allows us to evaluate 9 algorithms on 8 tasks using 8 seeds each (and additionally 15 ablation studies of IPBT on 4 tasks using 8 seeds each), giving both broad coverage and statistical power. The experiments in the paper are targeting our core interest and claim (creating a better hyperparameter optimization algorithm) rather than achieving state-of-the-art results on specific tasks, which the paper makes no claims about.
> > >
> > > > How is the performance when using a common computational budget in reinforcement learning such as a million environmental steps?
> > >
> > > As previously mentioned, the same computational budget was used for all algorithms in the experiments reported in the paper. In reinforcement learning experiments, 150M environmental steps were used by each algorithm (which is standard for these tasks in the literature [1]).
> > >
> > > > Are there any studies using restart and re-using weight information for hyperparameter tuning? If so, discussion with proper citations should be added.
> > >
> > > As described in Section 2.2, BG-PBT [1] introduced the idea of restarts into the PBT paradigm, with knowledge distillation as the weight re-use mechanism. Additionally, [2] was cited in Section 2.3 as an example of application of shrink-perturb (in that study, shrink-perturb is used in the PBT paradigm to re-use weights across architectures during architecture optimization - we will provide this extended description in the updated paper). Upon further literature search, we found that in [3], weight information was reused during Bayesian hyperparameter optimization - we will extend our discussion of relevant work with this citation.
> > >
> > > > Where are the evidences that BG-PBT requires more running time?
> > >
> > > As described in Sections 4.1 and 4.2, BG-PBT achieves results tied with IPBT when BG-PBT is run 4 times with different step sizes, and the best run is chosen. Since each run uses the same number of environment steps, running BG-PBT 4 times would require 4 times more computational resources. It can additionally be seen in Figure 4 that running BG-PBT without per-task step size tuning leads to significantly worse results than those of IPBT.
> > >
> > > ##### References:
> > >
> > > [1] Wan, Xingchen, et al. "Bayesian Generational Population-based Training." _International conference on automated machine learning_. PMLR, 2022.
> > >
> > > [2] Chebykin, Alexander, et al. "Shrink-Perturb Improves Architecture Mixing During Population Based Training for Neural Architecture Search." _26th European Conference on Artificial Intelligence, ECAI 2023_. IOS Press, 2023.
> > >
> > > [3] Mehta, Nikhil, et al. "Improving Hyperparameter Optimization with Checkpointed Model Weights." _European Conference on Computer Vision_. Cham: Springer Nature Switzerland, 2024.

---

### Official Review · Reviewer_TQZz · 2026-03-13

**Soundness:** 3
**Presentation:** 3
**Significance:** 3
**Originality:** 3
**Overall Recommendation:** 4
**Confidence:** 3

**Summary:**

This paper introduces Iterated Population Based Training (IPBT), a PBT variant that automatically adjusts the meta hyperparameter of the number of steps between hyperparameter update iteration. At each iteration, if performance stagnates, they generate new configurations and use the weights from the previous iteration to initialize the new iteration using shrink and perturb. Otherwise the method only resets the worst members of the population and let the others continue training. If doing a full reset, they increase the step size (in this case doubling). The paper evaluates IPBT on a set of classification and RL tasks, outperforming the baselines that are not tuned per task, and matching those that are.

**Compliance With Llm Reviewing Policy:**

Affirmed.

**Final Justification:**

Overall my concerns have been mostly addressed. I will maintain my score.

**Key Questions For Authors:**

- The pseudocode in the appendix was helpful in understanding the method, given it has several parts, but several key parts of your method are not explicitly shown, and just given a reference to the paper. Could you add the pseudocode for those as well?
- How were the hyperparameters for this method tuned? Were they tuned on just a subset of the tasks or all the tasks? If all the tasks, then it might be the case that these settings might not work out of the box on other tasks.

**Limitations:**

Yes

**Strengths And Weaknesses:**

Strengths:
- The method performs well overall, outperforming baselines that are not tuned per task, and approximately matching the best per task tuned baselines.
- The paper was well written and easy to follow
- The ablation study clearly shows the benefit of shrink-perturb and the step size adjustment.
- Method works well on both supervised learning and RL.

Weaknesses:
- Some design choices are still heuristics, e.g. hyperparameters for the method or restart threshold being 1 sd.
- It's a bit unclear to me if the hyperparameters found by the method would transfer to another seed. Because of the shrink-perturb based transfer, it could just be exploring/exploiting different weight initializations. If so, then maybe this is more of an optimization algorithm for the weights rather than the hyperparameters. Maybe rerun an experiment with the same hyperparameter schedules but different weight initializations.

---

> ### Author Rebuttal · Authors · 2026-03-30
>
> We thank the reviewer for the helpful feedback.
>
> > \[Key question 1:\] The pseudocode in the appendix was helpful in understanding the method, given it has several parts, but several key parts of your method are not explicitly shown, and just given a reference to the paper. Could you add the pseudocode for those as well?
>
> We agree with the reviewer that extending the general pseudocode with that of specific components would be helpful and we will add that to Appendix A.
>
> > \[Key question 2:\] How were the hyperparameters for this method tuned? Were they tuned on just a subset of the tasks or all the tasks?
>
> The method's hyperparameters were set based on a subset of the tasks (CIFAR-10/100, Humanoid, Hopper) using seeds different from those in the main experiments and ablations (see Appendix D for further details).
>
> > \[Weakness:\] It's a bit unclear to me if the hyperparameters found by the method would transfer to another seed. [...] Maybe rerun an experiment with the same hyperparameter schedules but different weight initializations.
>
> We have followed the reviewer's suggestion by conducting additional experiments on the CIFAR-10/100, and Fashion-MNIST datasets. For each of the 8 runs of IPBT on each dataset, we have extracted the best hyperparameter schedule produced by IPBT and used it to train models with different initializations and using different random seeds.
>
> We report the inter-quartile mean on test data, with the inter-quartile range in brackets:
>
> | Dataset       | IPBT                 | Replay optimal schedule |
> | ------------- | -------------------- | ----------------------- |
> | CIFAR-100     | 73.01 (72.60, 73.33) | 73.42 (73.11, 73.86)    |
> | CIFAR-10      | 92.75 (92.58, 92.89) | 92.99 (92.86, 93.16)    |
> | Fashion-MNIST | 94.95 (94.83, 95.07) | 95.02 (94.94, 95.12)    |
>
> It can be seen that the performance of a replay is as good as that of the original optimization run. We will include these results in the updated manuscript.

---

> > ### Author Rebuttal · Reviewer_TQZz · 2026-04-02
> >
> > This is encouraging, but I am a bit more curious about the results on RL, as I think they may be a bit more affected by the weight initialization that is found through IPBT. If you could rerun those with the optimal schedule found, that would be helpful. Also, just to confirm, you are taking only the hyperparameter schedules found through your procedure, and applying them to completely new random initializations (and not applying any of the population based reset mechanisms)?

---

> > > ### Author Response · Authors · 2026-04-08
> > >
> > > We agree with the reviewer that RL is generally less stable and more sensitive to weight initialization and the randomness present in the training procedure. Therefore, it is natural that training a population of networks improves performance and stability - but so does picking the best-performing schedule found by a non-population-based algorithm such as ASHA.
> > >
> > > To address the reviewer's question about the results for RL, we have repeated the schedule replay experiment (which is described in our previous comment) on the Humanoid task. The schedule produced by IPBT/ASHA in each of the 8 runs was used to train new models with new weight initializations and random seeds (which are matched for IPBT and ASHA, making the comparison fair).
> > >
> > > In the table below, inter-quartile means (of 8 runs) on test seeds are reported, with inter-quartile ranges in brackets.
> > >
> > > | Method | Original (8 runs; each run uses compute equivalent to training 8 networks) | Replay optimal schedule (8 runs; each run corresponds to training 1 network) |
> > > | ------- | --- | --- |
> > > | IPBT   |     9272.19 (8830.80, 9736.54) | 6939.43 (4372.02, 8887.38) |
> > > | ASHA   |     8241.37 (7240.89, 8940.58) | 6196.68 (4449.35, 7089.35) |
> > >
> > > It can be seen that both algorithms show worse performance and stability, which is in line with the inherent randomness and instability of RL that is not fully addressed by using an optimized schedule. Nonetheless, IPBT achieves higher IQM than ASHA not only in the original experiments but also when the optimized schedules are replayed, demonstrating that the improvement of IPBT over ASHA persists.
> > >
> > > We thank the reviewer for prompting us to run these experiments demonstrating that replaying the optimal schedule can be expected to give similar results in more stable settings such as image classification but diminished results in less stable settings such as RL. In such settings, it may be better to treat PBT variants as weight training methods rather than hyperparameter optimization methods, as the reviewer suggested and as is sometimes done in the literature [1].
> > >
> > > We will include these results and discussion in the updated manuscript.
> > >
> > > Regarding the reviewer's second question on applying population-based reset mechanisms during replay, we would like to clarify that only population-independent shrink-perturb has been applied on replay at the same points as it was during optimization.
> > >
> > > Logically, since the weights that were optimized by IPBT had shrink-perturb applied to them at specific points of the schedule, replaying that schedule requires replaying shrink-perturb. Since shrink-perturb is a cheap operation taking only ~0.5 seconds and not requiring a population, replaying it is effectively free. This is in contrast to BG-PBT, the runner-up PBT variant, that instead uses knowledge distillation, an expensive procedure that cannot be easily replayed.
> > >
> > > ##### References
> > >
> > > [1] Siqi Liu, Guy Lever, Zhe Wang, Josh Merel, S. M. Ali Eslami, Daniel Hennes, Wojciech M. Czarnecki, Yuval Tassa, Shayegan Omidshafiei, Abbas Abdolmaleki, Noah Y. Siegel, Leonard Hasenclever, Luke Marris, Saran Tunyasuvunakool, H. Francis Song, Markus Wulfmeier, Paul Muller, Tuomas Haarnoja, Brendan Tracey, Karl Tuyls, Thore Graepel, and Nicolas Heess. "From motor control to team play in simulated humanoid football." _Science Robotics_, 7(69), 2022.

---

### Official Review · Reviewer_GKUS · 2026-03-21

**Soundness:** 3
**Presentation:** 2
**Significance:** 3
**Originality:** 3
**Overall Recommendation:** 4
**Confidence:** 3

**Summary:**

This work introduces Iterated Population Based Training (IPBT), a new Hyperparameter Optimization (HPO) algorithm designed to improve upon standard Population Based Training (PBT). Standard PBT variants are sensitive to the "step size", while  IPBT addresses this by automatically adjusting the step size through a series of restarts and exponential increase. It also leverage shrink-perturb to do task-agnostic restart, improving search efficiency.

**Compliance With Llm Reviewing Policy:**

Affirmed.

**Final Justification:**

Will keep my score.

**Key Questions For Authors:**

Would it be possible to also generate a fixed recipe for repeated training?

**Limitations:**

Yes.

**Strengths And Weaknesses:**

**Strengths**

IPBT solves a major limitation of standard Population Based Training (PBT) by automatically adjusting the "step size".

The budget is practical: The algorithm is specifically designed for low-budget HPO (e.g., eight full training runs) and maintains the PBT advantage of running in parallel, taking approximately the same wall-clock time as a single training run

Achieving SOTA performance:  IPBT strongly outperforms previous PBT variants when their step sizes are untuned and matches or exceeds their performance even when those variants are manually tuned. It also surpasses other popular HPO algorithms like Random Search, ASHA, and SMAC3.

**Weaknesses**

This work introduce new Meta-Hyperparameters. While IPBT removes the need to tune the step size, it introduces new internal hyperparameters, such as the shrink-perturb constants (λ-shrink​  and γ-perturb) and stagnation detection parameters (t-patience and t interval). Although the authors claim these work well "out of the box," their specific influence on different tasks may require further study.

IPBT is designed as a black-box optimizer. If prior information about a specific task is available, specialized algorithms incorporating that knowledge might outperform it.

The online search can hardly get a fixed training recipe for repeated training tasks (in many industry settings).

---

> ### Author Rebuttal · Authors · 2026-03-30
>
> We thank the reviewer for the helpful feedback.
>
> We agree with the reviewer that introducing new meta-hyperparameters is not desirable, which is why the new hyperparameters of IPBT remain fixed across tasks. Good performance  of IPBT  shows that tuning these hyperparameters is not necessary, although we agree that further investigation of potential benefits of tuning them per task is an interesting avenue of future work - we will update Section 5 of the paper to mention this.
>
> > \[Key question:\] Would it be possible to also generate a fixed recipe for repeated training?
>
> To answer the reviewer's key question, we have conducted additional experiments on the CIFAR-10/100, and Fashion-MNIST datasets. For each of the 8 runs of IPBT on each dataset, we have extracted the best hyperparameter schedule produced by IPBT and used it to train models with different initializations and using different random seeds.
>
> We report the inter-quartile mean on test data, with the inter-quartile range in brackets:
>
> | Dataset       | IPBT                 | Replay optimal schedule |
> | ------------- | -------------------- | ----------------------- |
> | CIFAR-100     | 73.01 (72.60, 73.33) | 73.42 (73.11, 73.86)    |
> | CIFAR-10      | 92.75 (92.58, 92.89) | 92.99 (92.86, 93.16)    |
> | Fashion-MNIST | 94.95 (94.83, 95.07) | 95.02 (94.94, 95.12)    |
>
> It can be seen that the performance of a replay is as good as that of the original optimization run. We will include these results in the updated manuscript. To conclude, the answer to the reviewer's key question is: yes, the fixed recipes produced by IPBT can be used for repeated training.

---

> > ### Author Rebuttal · Reviewer_GKUS · 2026-04-05
> >
> > It is good to know IPBT could generate a fixed recipe with even better result. I will keep my score and vote for accept.
> >
> > My follow-up question is that, whether this works for more architectures, especially transformers.

---

> > > ### Author Response · Authors · 2026-04-08
> > >
> > > We are happy to hear that the reviewer's concerns have been addressed.
> > >
> > > > My follow-up question is that, whether this works for more architectures, especially transformers.
> > >
> > > In principle, IPBT is a general hyperparameter optimization algorithm that is agnostic to the underlying architecture. Empirical validation beyond convolutional networks (used in the image classification tasks) and MLPs (used in the reinforcement learning tasks) is an interesting avenue for future work (which we will mention in the updated Section 5).

---

### Decision · Program_Chairs · 2026-04-30

**Decision:**

Accept (regular)

**Comment:**

There has been some discrepancy among reviewers after the rebuttal phase. While some reviewers voted to accept the paper, another reviewer argued that it lacks originality, noting that similar ideas have been explored in different contexts and that the experiments do not include more recent architectures such as transformers.

While I agree that the contribution is somewhat incremental and that ideas such as weight inheritance or adaptive step sizes have been explored before, the novelty of the paper lies in investigating these ideas in the context of population-based training, which, to the best of my knowledge, has not been done before. I also agree that results on state-of-the-art architectures (e.g., transformer-based models) would be more convincing. However, the empirical evaluation appears solid, and the benchmarks are consistent with those commonly used in the literature. Overall, I agree with the other reviewers that these concerns do not justify rejecting the paper.